METHODS

# A statistical framework to identify cell types whose genetically regulated proportions are associated with complex diseases

Wei Liu[1], Wenxuan Deng[2], Ming Chen[2], Zihan Dong[2], Biqing Zhu[1], Zhaolong Yu[1], Daiwei Tang[2], Maor Sauler[3], Chen Lin[2], Louise V. Wain[4,5], Michael H. Cho[6,7], Naftali Kaminski[3], Hongyu Zhao[1,2] *

1 Program of Computational Biology and Bioinformatics, Yale University, New Haven, Connecticut, United States of America, 2 Department of Biostatistics, Yale School of Public Health, Yale University, New Haven, Connecticut, United States of America, 3 Pulmonary, Critical Care and Sleep Medicine, Yale School of Medicine, Yale University, New Haven, Connecticut, United States of America, 4 Department of Health Sciences, University of Leicester, Leicester, United Kingdom, 5 National Institute for Health Research, Leicester Respiratory Biomedical Research Centre, Glenfield Hospital, Leicester, United Kingdom, 6 Channing Division of Network Medicine, Brigham and Women's Hospital, Harvard Medical School, Boston, Massachusetts, United States of America, 7 Pulmonary and Critical Care Medicine, Brigham and Women's Hospital, Harvard Medical School, Boston, Massachusetts, United States of America

☯ These authors contributed equally to this work.
* hongyu.zhao@yale.edu

**Data Availability Statement:** All data used in this manuscript can be requested from the following URLs: Human cell landscape: GEO GSE134355 https://www.ncbi.nlm.nih.gov/geo/query/acc.cgi?

## Abstract

Finding disease-relevant tissues and cell types can facilitate the identification and investigation of functional genes and variants. In particular, cell type proportions can serve as potential disease predictive biomarkers. In this manuscript, we introduce a novel statistical framework, cell-type Wide Association Study (cWAS), that integrates genetic data with transcriptomics data to identify cell types whose genetically regulated proportions (GRPs) are disease/trait-associated. On simulated and real GWAS data, cWAS showed good statistical power with newly identified significant GRP associations in disease-associated tissues. More specifically, GRPs of endothelial and myofibroblasts in lung tissue were associated with Idiopathic Pulmonary Fibrosis and Chronic Obstructive Pulmonary Disease, respectively. For breast cancer, the GRP of blood CD8[+] T cells was negatively associated with breast cancer (BC) risk as well as survival. Overall, cWAS is a powerful tool to reveal cell types associated with complex diseases mediated by GRPs.

## Author summary

Cell type proportions such as T cell proportions have been found to be potential disease progression indicator especially for cancer patients. However, cell type proportion changes can result from disease status, thus making it difficult to know whether those changed proportions are due to disease progression or the cause of disease status. Genetic components of cell type proportions, however, can potentially help to identify cell type proportions leading to different disease statuses. Here we introduce a novel statistical

acc=GSE134355; CNGBdb CNP0000325 https://db.
cngb.org/search/project/CNP0000325/ GTEx data:
dbGaP Accession phs000424.v8.p2Roadmap
Epigenomics project: http://egg2.wustl.edu/
roadmap/data/byFileType/signal/consolidated/ BC
summary stats: https://bcac.ccge.medschl.cam.ac.
uk/bcacdata/oncoarray/oncoarray-and-combined-
summary-result/gwas-summary-associations-
breast-cancer-risk-2020/ COPD summary stats:
https://pubmed.ncbi.nlm.nih.gov/24621683/;
https://pubmed.ncbi.nlm.nih.gov/30804561/ IPF
summary stats: https://github.com/genomicsITER/
PFgenetics MAGMA: https://github.com/Kyoko-
wtnb/FUMA_scRNA_data.

**Funding:** Part of this work was supported by
funding from NIH grant R01GM134005 and NSF
grant DMS1902903 to H.Z., NHLBI grants
R01HL127349, R01HL141852, U01HL145567,
UH2HL123886 to N.K. and a generous gift (non-
restricted financial support with no grant number)
from Three Lakes Partners to N.K. M.H.C. was
supported by NHLBI grants R01HL135142, R01
HL137927, R01 HL089856, R01 HL147148. L.V.
W. was supported by the NIHR Leicester
Biomedical Research Centre; the views expressed
are those of the authors and not necessarily those
of the NHS, the NIHR or the Department of Health.
The funders had no role in study design, data
collection and analysis, decision to publish, or
preparation of the manuscript.

**Competing interests:** I have read the journal's
policy and the authors of this manuscript have the
following competing interests: N.K. served as a
consultant to Boehringer Ingelheim, Third Rock,
Pliant, Samumed, NuMedii, Theravance, LifeMax,
Three Lake Partners, Optikira, Astra Zeneca,
Augmanity over the last 3 years, reports Equity in
Pliant and a grant from Veracyte and Boehringer
Ingelheim and non-financial support from MiRagen
and Astra Zeneca. N.K. as IP on novel biomarkers
and therapeutics in IPF licensed to Biotech. M.H.C.
has received grant support from GSK and Bayer,
consulting or speaking fees from Genentech,
AstraZeneca, and Illumina. L.V.M. holds a GSK/
British Lung Foundation Chair in Respiratory
Research.

framework, cell-type Wide Association Study (cWAS), that integrates genetic data with transcriptomics data to identify cell types whose genetically regulated proportions (GRPs) are disease/trait-associated. In simulated data, cWAS showed a high statistical power in identifying disease-associated cell type associations with a well-controlled type-I error rate. Applying cWAS to breast cancer data, we found that blood CD8+ T cells may serve as the protective factor against breast cancer, i.e. high CD8+ T cell proportions lead to lower breast cancer risks and better prognostic condition. Overall, cWAS is a powerful tool to help identify disease-associated cell type proportions and potentially help in clinical research and practices.

## Introduction

Despite the great success of genome-wide association studies (GWAS), it has been challenging to identify disease-causing genes and variants. To better design functional studies of GWAS-implicated SNPs, it is important to identify tissues and cell types most relevant to a disease. Several statistical approaches have been developed for this purpose [1–3]. In general, these methods aim to detect statistically significant overlap between GWAS signals and annotated functional regions in specific tissues and cell types, where the annotated functional regions are curated from other data sources, such as ENCODE [4] and Roadmap Epigenomics [5] data and single cell data. Although such analyses have led to novel insights on disease mechanisms [6–9], the cell types associated with the majority of genomic regions remain to be discovered.

Several studies have found that the proportions of cell types are not only associated with disease incidence [10,11] but also disease prognosis [12,13]. Single cell RNA-seq (scRNA-seq) technologies have been used to identify cell type proportions that impact human diseases and traits [14]. However, several intrinsic characteristics of single cell data make disease-cell type proportion association analysis challenging. First, high expense and technical noise (e.g., high sparsity of gene expression) limit the number of samples analyzed and quality of cell type composition estimation, leading to low power in association analysis. Second, cell type compositions measured in single cell experiments are highly dependent on the biopsy samples and do not necessarily reflect the true cell type compositions in the corresponding tissue [15]. Instead of directly calculating cell type proportions from scRNA-seq data, cell type proportions can also be inferred through deconvolution of bulk RNA-sequencing (RNA-seq) data available with larger sample sizes. Many computational methods have been developed to estimate cell type proportions in bulk RNA-seq data using cell type-specific gene expression signatures derived from either microarray or scRNA-seq reference [15]. Compared with biopsy samples in single cell analyses, tissue samples for bulk analysis might better represent the original cell type compositions [10,15].

For both single cell and bulk data, cell type proportions can be affected by various factors including disease status and treatment effects. Consequently, the observed cell type proportion differences between disease and healthy individuals might be the outcome of the disease and environmental factors instead of disease causes.

Unlike assayed gene expression levels, genotypes are less likely to be affected by confounding factors and reverse causation. The same idea underlies Mendelian randomization methods to infer causal factors for different traits [16–18]. In this paper, we examined genetically regulated proportions (GRPs) of cell types. We note that cell type proportions are heritable [13,19], suggesting the feasibility of inferring cell type proportions based on genotypes. Cell type proportions can vary substantially in patients with different diseases [11]. We introduce a new

framework, cell-type Wide Association Study (cWAS), to consider the GRPs of cell types as contributors to human disease. Through simulation studies and real data analyses across 56 traits in 36 tissues, cWAS showed higher statistical power in identifying disease-cell type proportion associations than commonly used cell-disease association identification approaches such as LD score regression [1] and FUMA [3]. In summary, cWAS offers a novel way to understand human diseases in a cell-type specific manner.

## Results

### Model summary

We propose a statistical framework to identify cell types whose GRPs are associated with diseases. The framework consists of two parts (**Fig 1**). First, under the assumption that there exist signature genes signifying specific cell types (consistent with previous methods [19,20]), we infer GRPs of cell types through deconvolution of the imputed tissue-specific gene expression levels based on SNP genotypes from eQTL data. Second, we combine the inferred GRPs with disease phenotype information to identify cell-type proportion associations with disease phenotypes.

In the first step, we build tissue-specific gene expression imputation models using the elastic net, similar to previous Transcriptome-Wide Association Study (TWAS) methods [21–23]. With the imputation weights $\hat{\beta}_{gt}$, we estimate genetically regulated tissue-level gene expression for gene $g$ in tissue $t$ as $\hat{B}_{gt} = X_g \hat{\beta}_{gt}$, where $X_g$ is the genotype matrix of cis-SNPs around gene $g$. With pre-specified cell-type specific gene expression levels for signature genes, we deconvolute the genetically imputed tissue-level expression data through the following model:

$$\hat{B}_t = \hat{F}_t S_t^T,$$

where $\hat{B}_t$ is the imputed gene expression level matrix for all signature genes in tissue $t$, $S_t \in R^{G \times C}$ is the cell-type specific gene expression level matrix in tissue $t$ for $G$ signature genes across $C$ cell types, and $\hat{F}_t$ is the estimated GRPs for all cell types in tissue $t$. For a specific cell type $c$, we assess its GRP association with phenotype $Y$ using the following model:

$$Y = \sum_t \sum_c \hat{F}_{(.,c),t} \gamma_{c,t} + \eta,$$

where $\gamma_{c,t}$ is the effect of GRP for cell type $c$ of tissue $t$ on the trait, one element of the effect vector $\gamma_t$ of GRP of tissue $t$; and $\eta$ includes both genetic effects not mediated by genetically regulated cell type proportions and random noises. $\hat{F}_{(.,c),t}$ is the estimated GRPs of a cell type $c$ in tissue $t$, which is the $c$th column of the $\hat{F}_t$ matrix. However, individual-level genotype data are not always available for GWAS, which makes the direct estimation of $\gamma_{c,t}$ from the above two-step procedure infeasible. With only summary statistics available and assuming the effects $\gamma_{c,t}$ of all tissue $t$ and cell type $c$ are independent of each other, we propose to use the following approach to assessing the marginal association between GRPs of a cell type $c$ in tissue $t$ and traits

$$z_{c,t} \approx \sum_p sd(X_p) z_p \hat{\beta}_{t,p} S_t (S_t^T S_t)_{.,c}^{-1} / sd(\hat{F}_{(.,c),t}),$$

where $sd(X_p)$ is the genotype standard deviation of SNP $p$, calculated from a reference panel; $z_p$ is the GWAS z score for SNP $p$; $\hat{\beta}_{t,p}$ is the imputed tissue-level gene expression vector of SNP $p$ across $G$ signature genes in tissue $t$; $(.)_{.,c}$ stands for the $c$th column vector of the corresponding matrix; and $sd(\hat{F}_c)$ is the standard deviation of genetically regulated cell type proportions for

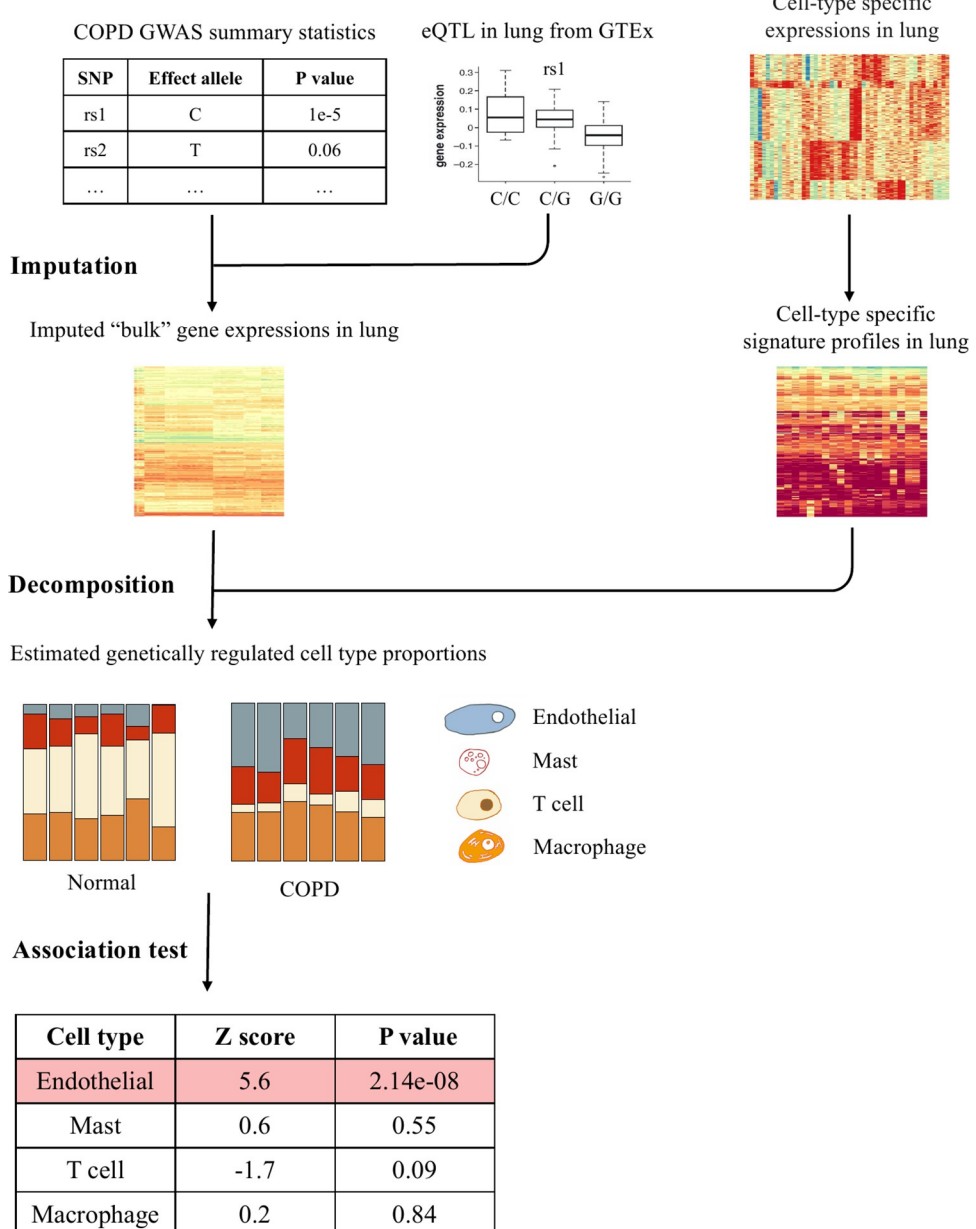

**Fig 1. The schematic framework of cWAS.** Bulk gene expression levels are firstly imputed based on each individual's genotypes. Combined with a signature gene expression matrix for different cell types, imputed gene expression data for each tissue are used to infer cell type proportions. Comparing genetically regulated cell type proportions in case and control individual groups, cWAS can identify cell types whose genetic-regulated proportions are associated with a trait of interest.

cell type $c$. Overall, cWAS assumes that the signature gene expression levels are similar across samples and the cell-type specific genetic effects on gene expression are mediated by regulating cell type proportion(s). Due to the limited power for identifying and estimating trans-eQTL effects, cWAS only considers cis-eQTL effects. cWAS takes the GWAS summary statistics as the input, which provides an indirect way of estimating cell-type GRP associations with diseases and does not require individual-level data. Corresponding tissue-level multiple-test

correction was applied for test statistics in each tissue respectively. Overall, the test statistic $z_c$ is similar to the GWAS burden test statistic which considers the signature gene matrix and tissue-level eQTL effects as weights to quantify the overall genetic effects mediated by regulating cell type proportions on traits of interests, while $sd(X_p)$ and $sd(\hat{F}_c)$ are used to normalize the genotype variance and imputed cell type proportion differences. More model details are presented in the methods section, and the cWAS framework for GWAS summary statistics is available at https://github.com/vivid-/cWAS.

## Simulation studies

To evaluate cWAS performance in identifying cell type proportions associated with a disease, we considered several simulation settings (**Methods**). We simulated disease phenotypes based on genetically predicted proportions of M1 macrophages in whole blood, using 10,000 individuals randomly sampled from UK Biobank [24]. Under moderate heritability settings, where genetically regulated cell type proportions explain 1% to 9% of the phenotype variances, cWAS had at least 98% power to identify M1 macrophages' association with the phenotype when all signature genes were known and used (**Fig 2A,** the purple dashed line). Nevertheless, the strong proportion dependency between cell types can potentially lead to the false identification of cell types whose proportions are highly correlated with the true signal cell type. Correlated cell types will be more likely to be identified as disease-associated together (r = 8e-2, p = 1.8e-2, **S1 text**). To mitigate this situation, we proposed a permutation-like test whose statistical power is comparable to the z-statistic test used here but with better controlled false identification rates when cell types are highly correlated. Due to high computation burden of permutation test and similar performances (**S7 Fig**), we are using the z-statistic test here (**S1 Text**). When only focused on the most significant cell type, cWAS still demonstrated a significant improvement. M1 macrophage was identified as the most significant cell type in at least 70% of 600 Monte Carlo replicates (**Fig 2B**) when heritability was 4% or higher, and the effect of M1 macrophages identified by cWAS had the same direction as that simulated in at least 90% of 600 replicates, while FUMA only identified macrophages as the significant cell type in around 15% of 600 replicates (**Table A in S1 Table**). When we simulated phenotypes independent of cell type proportions in the whole blood tissue, cWAS had a well-controlled type I error rate (**Fig 2C**).

It is critical for cWAS to have reliable cell type specific gene expression signatures. Many cell-type deconvolution methods also depend on the accurate curation of the signature matrix, such as those from microarray data of known cell types (like the LM22 matrix used in CIBERSORT [20]. However, in many cases, we have to derive a signature matrix from single-cell data, which are usually highly sparse and only include cell type-specific expression levels of a subset of signature genes. Consequently, the signature genes curated from single-cell data may be incomplete compared to those from more informative data sources, such as RNA-seq assayed in known cell types. To evaluate the impact of incomplete genes in the signature matrix, we considered using a subset (50%-90%) of signature genes in cWAS. When only half of the signature genes were used, there was a significant drop in statistical power although the type I error was still well-controlled (**Fig 2D**). With an increasing proportion of signature genes used, there was improved power in identifying associated cell types (**Fig 2A**). Besides, based on our simulation studies, the eQTL study sample size has a more significant effect on the cWAS statistical power than the GWAS sample size (**S1 Text**). Compared to GWAS sample size (p = 0.16), the eQTL sample size (p = 3.6e-7) has more significant effects on the statistical power. This indicates that the imputation weights/models for bulk-level gene expression are more likely to limit the statistical power of cWAS models (**S10 Fig**).

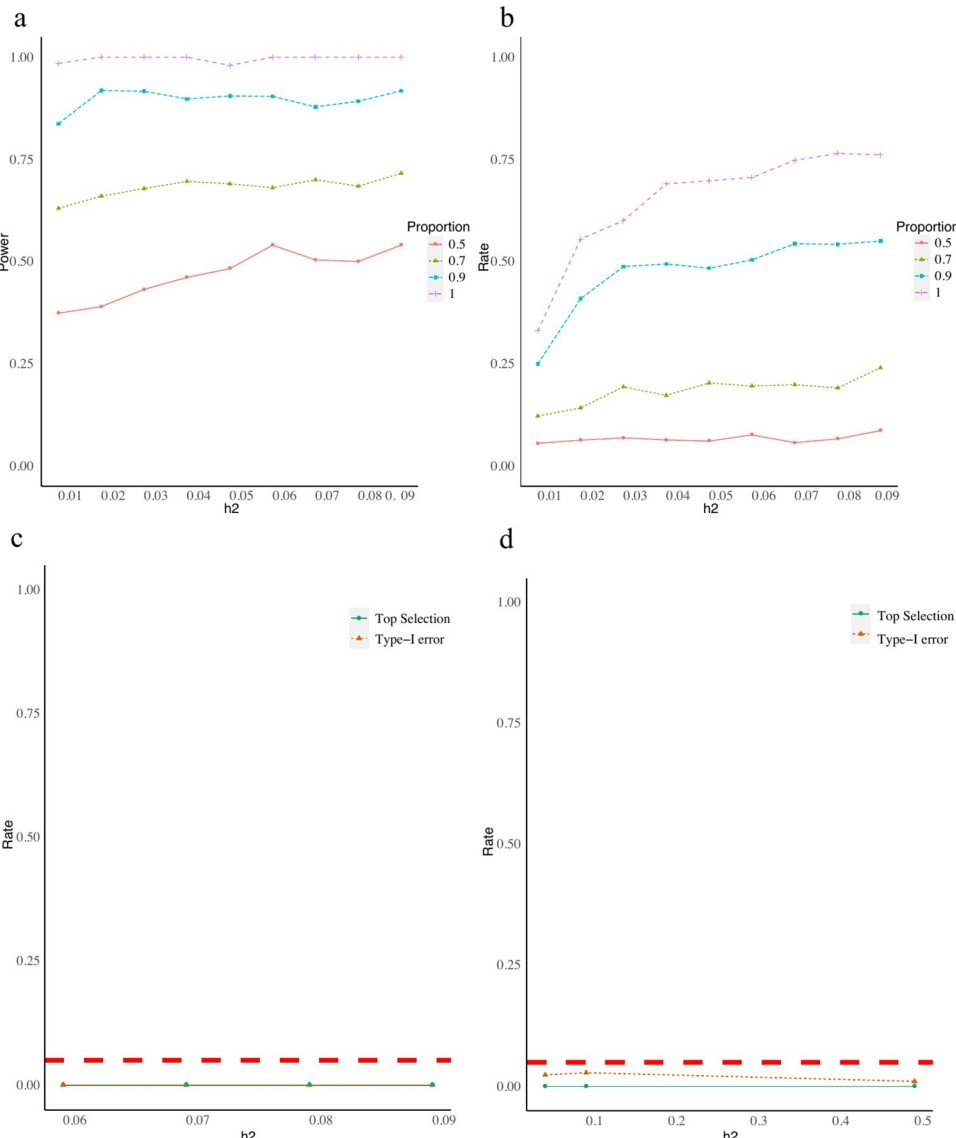

**Fig 2. cWAS performances in simulation studies.** Different colors indicate different proportions (0.5, 0.7, 0.9, and 1) of signature genes used in the cWAS test. The phenotypic variance explained by the genetically regulated cell type proportions (M1 macrophages) ranges from 0.01 to 0.09 for panels a and b, respectively. a) Each line represents the percentage of simulations where cWAS identified the M1 macrophages as associated with simulated phenotypes. b) Each line shows the proportion of times that the M1 macrophage was identified as the most significant cell type (top selection rate) whose proportion was associated with simulated phenotypes with different proportions of known signature genes. For panels c and d, we simulated phenotypes based on the genetically regulated proportion of basal cells in lung tissue with heritability being 0.05, 0.1, or 0.5. c) No cell type proportion in whole blood is associated with simulated phenotypes in this setting, and the test results shown here are those for whole blood genetically regulated cell type proportions. The simulated phenotypes are simulated as associated with genetically regulated cell type proportions in lung tissue. All signature genes in whole blood are known when conducting the cWAS test. The red dashed line indicates the 5% type I error. The green line indicates the proportion of simulations where any cell type in whole blood was selected as associated with the simulated disease status, the orange line indicates the proportion of simulations where M1 macrophages were selected as associated with the simulated disease. d) the same setting as that for c) and only 50% of signature genes in whole blood are known.

## Trait-tissue association patterns

To further study disease-cell type proportion associations, we applied cWAS to GWAS summary data from 56 traits (**Table B in S1 Table**, including autoimmune diseases, psychiatric disorders, and other traits like lipids and height) together with scRNA-seq data from the Human Cell Landscape (HCL) [25]. We identified trait-associated cell types in 21 adult non-brain tissues and 13 fetal brain tissues (**Table C in S1 Table**) using gene expression imputation models for curated signature genes (**Methods, S1 Fig**). Consistent with findings from other methods, we found that the most significant cell types are usually present in the trait-associated tissues [1, 26] (**Figs 3A and S2**) supporting the validity of cWAS, e.g., oligodendrocytes from fetal brain amygdala for autism spectrum disorder (ASD) (p = 3.0e-3), myeloid progenitor cells from whole blood for Crohn's disease (p = 3.6e-5), and endothelial cells from tibial artery for resting heart rate (HR) (p = 4.0e-9). Several traits showed global cell type proportion associations across multiple tissues, e.g., height and body mass index (BMI). Notably, cWAS identified many cell type-trait associations in unexpected tissues. Many of them are immune cells, for example, neutrophil cells in fetal brain frontal cortex are associated with systemic lupus erythematosus (SLE) (p = 5.8e-4), and macrophages from subcutaneous adipose and neutrophils from the left ventricle of the heart are associated with anxiety disorders (ADIS) (p = 7.4e-3 and 1.6e-3, respectively).

Since several cell types (**Table D in S1 Table**), especially immune cells, are present in multiple adult tissues, we further investigated whether those identified disease-associated immune cell types above are due to true biological processes or false positives by studying tissue-tissue correlations based on shared cell types' associations with traits (**Methods**). Compared to biologically unrelated tissue pairs, the results showed a higher correlation among tissues with similar biological functions (**Fig 3B**), such as artery tissues (tibial artery, coronary artery, and aorta artery), heart tissues (left heart ventricle and heart atrial appendage), and esophagus tissues (esophagus muscularis and esophagus mucosa). This finding suggests that cell types are more likely to be identified as trait-associated in disease-related tissues even though the same cell types may exist in multiple tissues. Also, to evaluate the consistency between cWAS and other GWAS downstream analysis methods, we also applied heritability enrichment tools and GWAS colocalization tool to investigate cell type and trait associations (see "Comparisons between cWAS, MAGMA gene analysis, LD score regression, and signal colocalization methods" in the **S1 Text**). Overall, cWAS results showed a high concordance with results from other methods.

We also evaluated correlations among traits based on their associations with different cell types across 21 adult non-brain tissues and 13 fetal brain tissues, respectively (**Table E in S1 Table**). In 21 adult non-brain tissues, we identified high correlations among many traits, e.g., autoimmune diseases including eczema and Systemic Lupus Erythematosus (SLE); lipid traits like total cholesterol (TC), low-density lipoprotein cholesterol (LDL), and triglycerides (TG) (**Fig 4A**). Brain tissue-associated traits have higher correlations based on estimates using fetal brain tissues (**Fig 4B**) compared to those from adult non-brain tissues. For example, Alzheimer's disease (AD) is clustered with autoimmune-related traits in adult non-brain tissues, whereas it is correlated with psychiatric traits like bipolar disorders (BD) and attention-deficit disorder (ADHD) in fetal brain tissues. For some other traits, their correlations in 13 fetal brain tissues were similar to those identified in adult non-brain tissues. For example, a positive correlation between ASD and ADHD was observed for both adult tissues (r = 0.33, p = 1.4e-7) and fetal brain tissues (r = 0.55, p = 7.9e-16). Moreover, we observed correlations in different directions between fetal brain tissues and adult non-brain tissues. For example, smoking initiation (SmkInit) and asthma had a positive

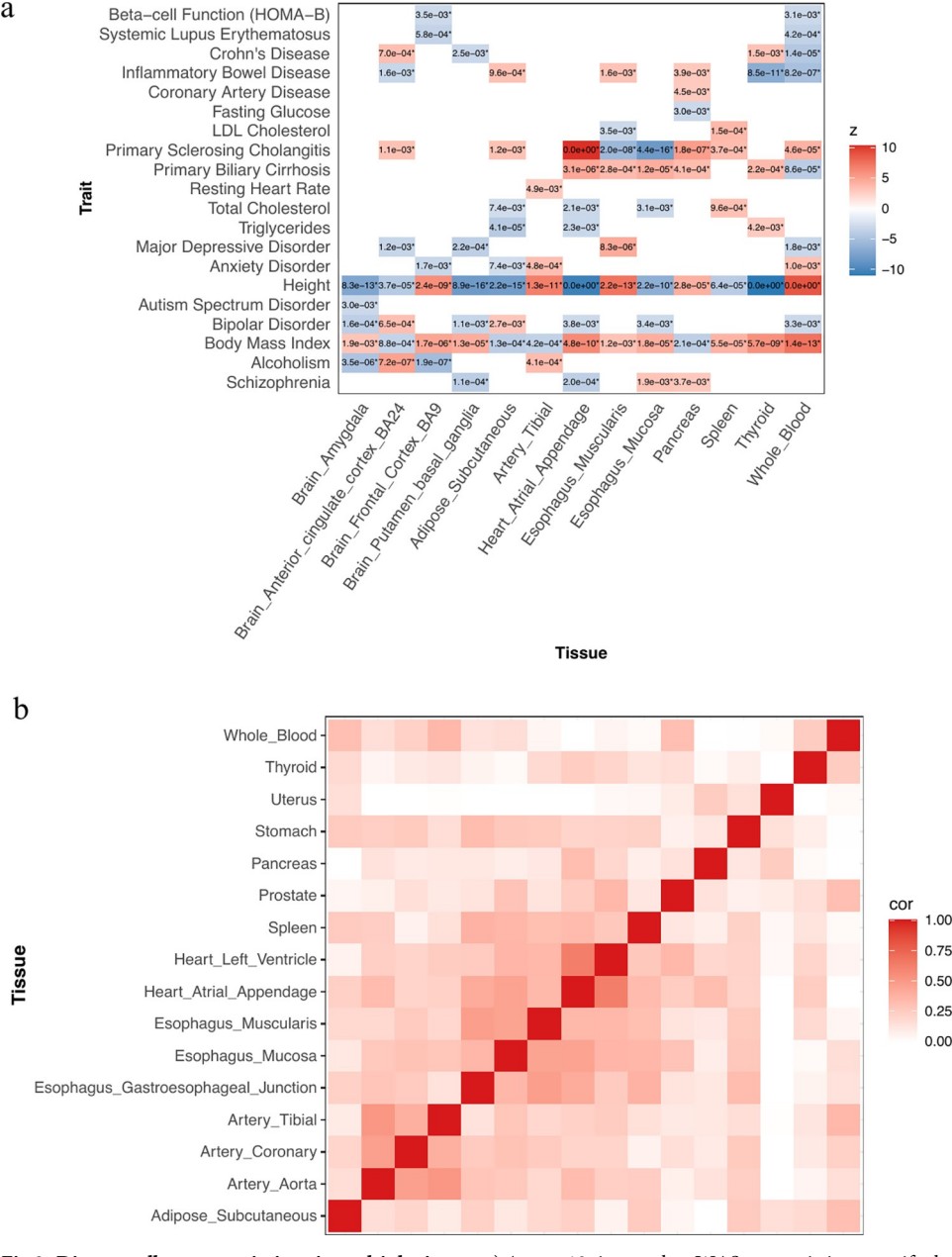

**Fig 3. Disease-cell type associations in multiple tissues.** a) Across 12 tissues, the cWAS test statistics quantify the associations between genetically regulated cell type proportions and diseases. If there is no cell type significantly associated with a disease after Bonferroni correction, the corresponding entry is blank. The number in each block indicates the p-value of the most significant association between the cell type proportion of the corresponding tissue and the disease. b) For any tissue pair, we only considered shared cell types and treated their proportion associations across 56 traits. Tissue-tissue correlations were calculated based on the cell type-disease associations for the shared cell types. The darker color indicates a higher significance level.

correlation in fetal brain tissues (r = 0.35, p = 1.1e-6) but a negative correlation in adult non-brain tissues (r = -0.33, p = 1.2e-7). The associations identified between asthma and neuronal cells in fetal brain tissues may be supported by previous findings linking neural pathways to allergic inflammation in lungs [27, 28].

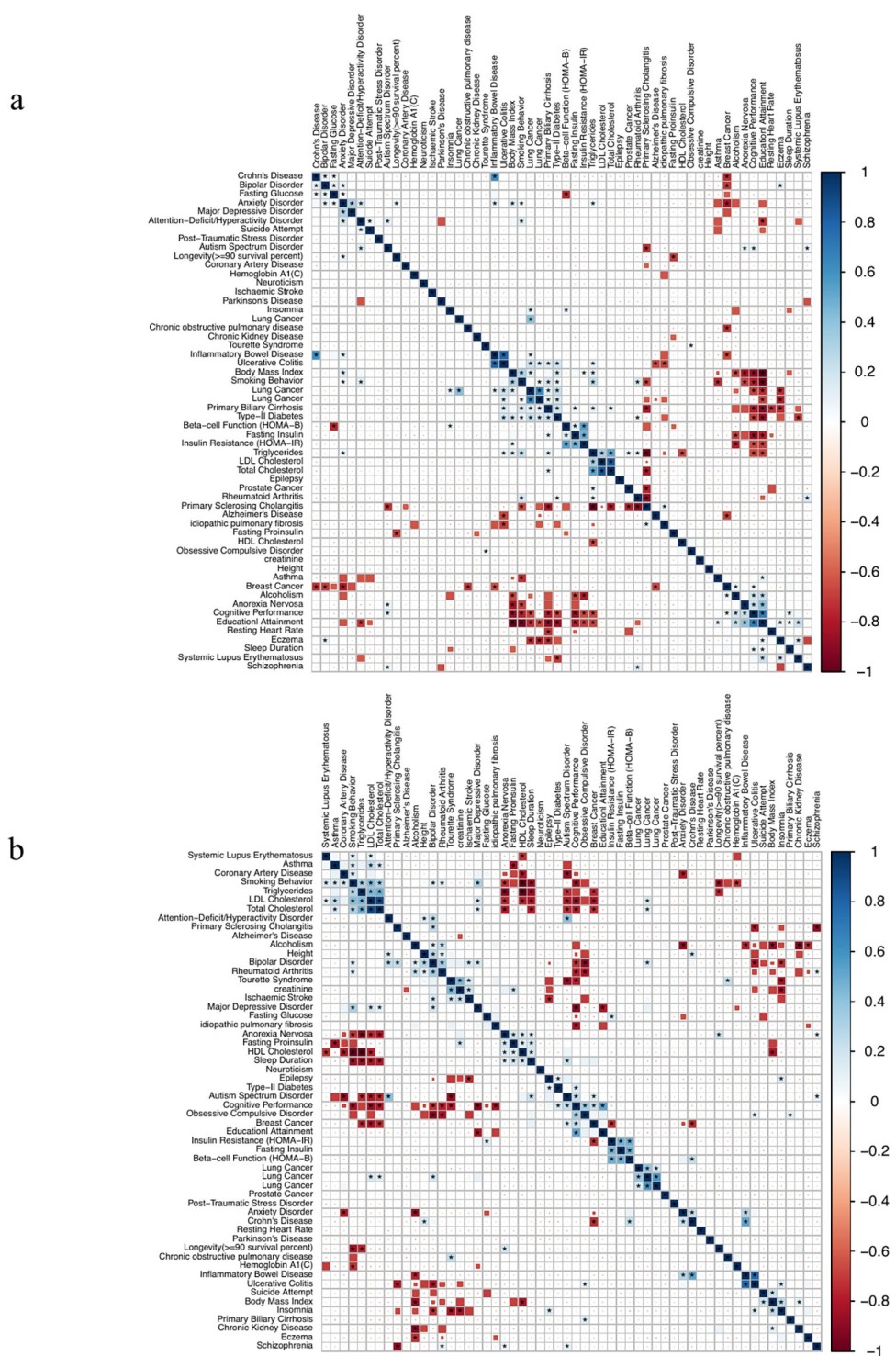

**Fig 4. Trait-trait correlation.** Different colors indicate the correlation level and the stars indicate the significant correlations after Bonferroni correction. a) Trait-trait correlation calculated from cell-disease associations in adult non-brain tissues. b) Trait-trait correlation based on cell-disease associations in fetal brain tissues.

## Breast cancer and CD8$^+$ T cells

To further examine the potential utility of cWAS using specific datasets, we applied cWAS to identify cell types for breast cancer and two lung diseases. For breast cancer (BC), we used European breast cancer GWAS summary data [29] (n = 228,951, n_case = 122,977, n_control = 105,974). In whole blood, we identified a significant negative association between GRPs of CD8$^+$ T cells and BC risk (**Fig 5A**) (p = 5.0e-4) using the published signature gene expression matrix LM22 [15,20].

To explore potential biological and clinical implications of this result, we imputed genetically regulated cell type proportions in whole blood for subjects with European ancestry in The Cancer Genome Atlas (TCGA) project who were diagnosed with BC (TCGA-BRCA) [30] (see **Methods**). We found that basal breast cancer patients with higher imputed CD8$^+$ T cell proportions had an overall better survival (**Fig 5B**, $p = 0.085$). Results were similar but significant ($p = 0.034$) for luminal B breast cancer patients (**Fig 5C**). We also considered an alternative approach to evaluating cell-type specific expression patterns of BC-associated genes identified using epigenetic annotations and genetic signals (T-GEN [31]). BC-associated genes showed no significant expression enrichment in any cell type of whole blood other than a significant depletion in dividing NK T cells (fold-change = 0.79, p = 1.6e-8) (**S3A Fig**). Furthermore, BC-associated genes identified by T-GEN did not show significantly higher expression levels in T cells or any other cell types (**S3B Fig**).

To further validate our results, we studied BC-cell type proportion association using another cell type proportion decomposition approach [32]. In this case, the cell type proportion association result was based on the directly measured tumor tissue transcriptome data from TCGA-BRCA. We found a similar protective effect of the CD8$^+$ T cell proportion ($p = 0.013$) in basal breast cancer patients (**S4A Fig**), but not in luminal breast cancer patients (**S4B and S4C Fig**).

## Lung diseases and lung tissue

Using single cell data [33] with better quality than HCL data to identify cell types with small proportions, we performed cWAS analysis for two lung diseases, idiopathic pulmonary fibrosis (IPF, n = 24,589, n_case = 4,124, n_control = 20,465) [34] and chronic obstructive pulmonary disease (COPD, n = 5,346, n_case = 2,812, n_control = 2,534) [35]. In IPF, a higher predicted proportion of myofibroblast in lung tissue was associated with an increased risk of developing the disease (p = 5.3e-4, **Fig 6A**), consistent with the accumulation of myofibroblasts observed in IPF patients [36]. We also observed a negative association of fibroblast proportions in the development of IPF (p = 3.5e-2), which is consistent with aberrant fibroblast-to-myofibroblast [37] differentiation and fibroblast degeneration and myofibroblast proliferation [38] in IPF.

To further evaluate cell type associations with IPF, we investigated the cell type expression pattern of IPF dysregulated genes by conventional transcriptomics analysis. Using differentially expressed genes from the published [39] RNA-seq data of lung tissue in IPF patients (*n* = 36) and non-disease individuals (*n* = 19), we found that upregulated genes in IPF patients were significantly enriched in myofibroblasts (fold change = 1.3, p = 1.4e-3, **Fig 6B**). However, genes differentially expressed in myofibroblasts can result either from genetic effects or disease status. We further analyzed the cell type signal based on genetic information using IPF GWAS summary statistics. Applying MAGMA (implemented in FUMA, see URLs) to the IPF GWAS results (**S5A Fig**), we found marginal evidence of enriched genetic signals in the fibroblasts of lung tissue (p = 7.5e-2), although not statistically significant. IPF-associated genes identified by T-GEN [31] did not show significant enrichment in any cell type of lung. Therefore, though neither was significant after Bonferroni correction, both transcriptomic and gene-set based

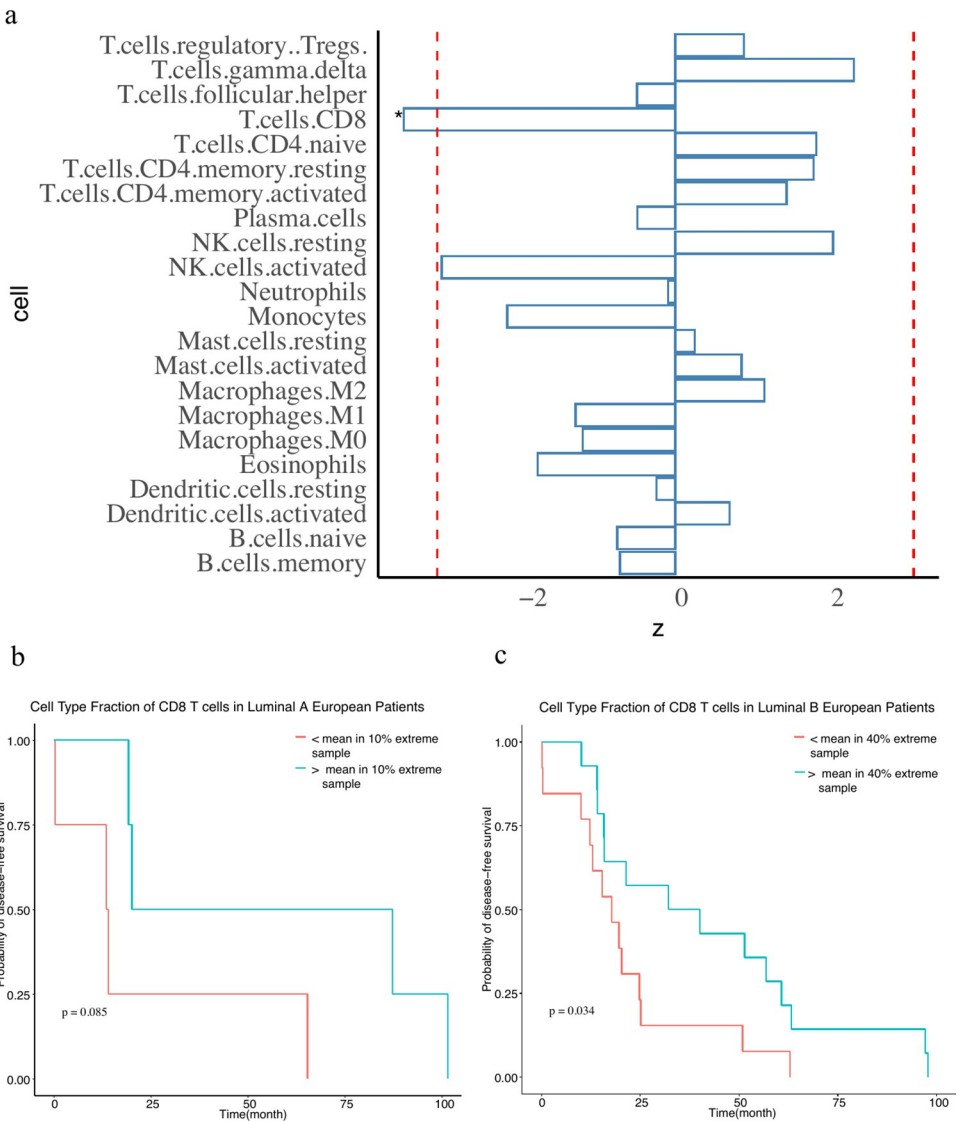

**Fig 5. CD8⁺ T cells in breast cancer.** a) cWAS results of breast cancer in whole blood. The x axis is the z score of the cell type-disease association from cWAS. Negative z scores indicate negative associations between cell type proportions and diseases. The red line indicates the significance threshold (0.05) after Bonferroni correction. The star indicates significant cell types after Bonferroni correction. b) and c) show survival analysis results in breast cancer patients of TCGA. b) Considering the white basal patients with top 10% and low 10% of genetically regulated cell type proportions of CD8⁺ T cells, and the survival patterns were compared between patients in these two groups. c) shows the results of a similar analysis in white Luminal B breast cancer patients considering patients with top 40% and bottom 40% of genetically regulated proportions of CD8⁺ T cells.

genetic analyses suggest the importance of myofibroblasts and fibroblasts consistent with cWAS.

For COPD, cWAS found higher GRPs of endothelial cells increased disease risk (p = 2.1e-4, **Fig 6C**). To further investigate the association, we applied cWAS in another GWAS [32] of larger sample size with a signature matrix having more refined cell types (**Methods**). One specific endothelial cell type, vascular endothelial capillary A, was positively associated (p = 3.9e-4) with COPD based on results from another GWAS (N = 257,811) [40]. Upregulated genes in COPD patient lung tissue [39] were also enriched in endothelial cells (fold change = 1.4, p-

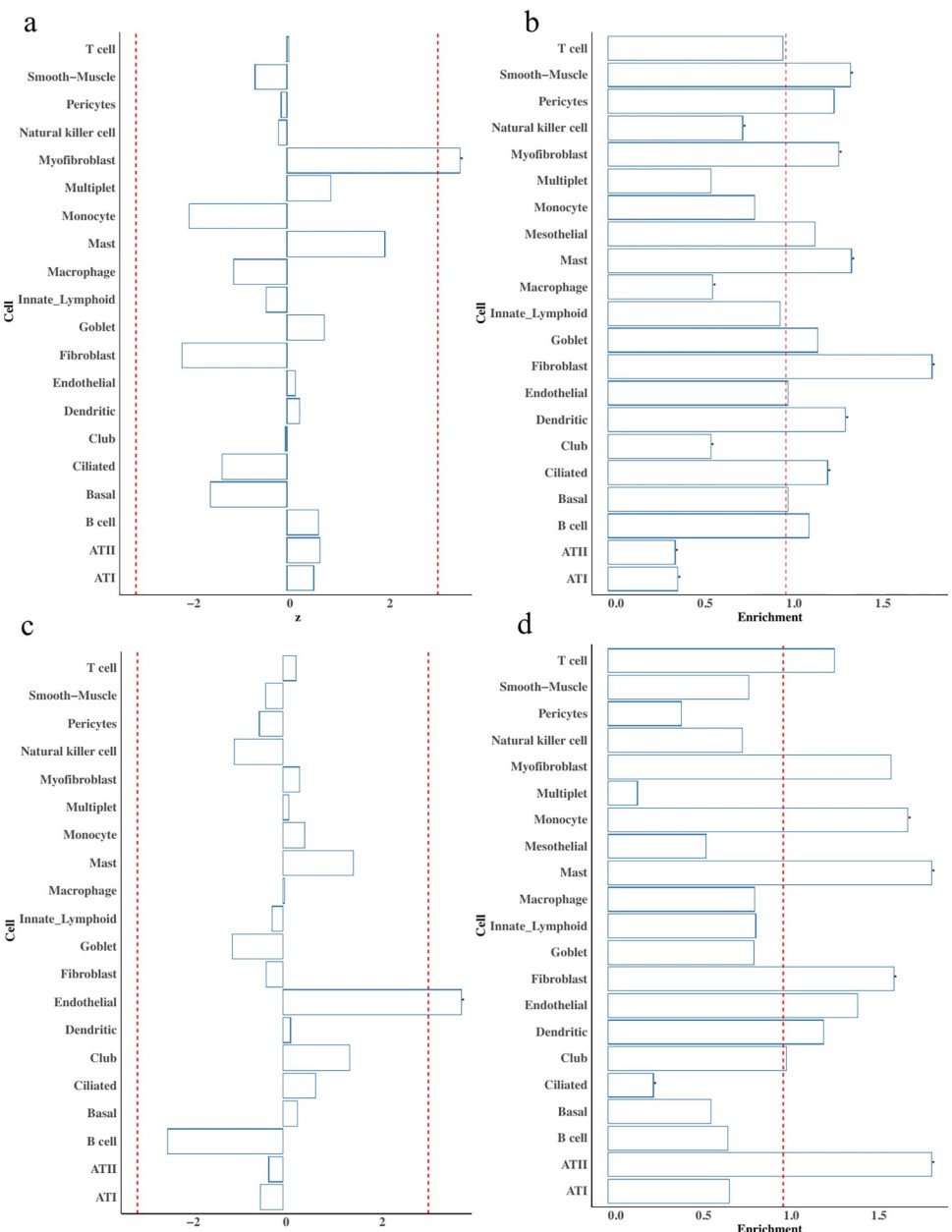

**Fig 6. cWAS association results of IPF and COPD in lung.** For a) and c), the red line indicates the significance threshold (0.05) after Bonferroni correction. For all figures, stars indicate significant cell types after Bonferroni correction. a) cWAS results of IPF in lung tissue. The x axis is the z score of the cell type-disease association from cWAS. Negative z scores indicate a negative association between cell type proportions and the disease. b) Cell-type specific expression enrichment pattern of upregulated genes in IPF patients. c) cWAS results of COPD in lung tissue. d) Cell-type specific expression enrichment pattern of upregulated genes in COPD patients.

value = 1.4e-2) (**Fig 6D**). Similar to IPF analysis, we also investigated COPD genetic signal enrichment using MAGMA on mouse lung data (no human lung data available in FUMA, **Table F in S1 Table**). There was marginal evidence of signal enrichment in endothelial cells in the lung tissue (p = 8e-2, **S Fig 5B**, statistically not significant) and lung vasculature (p = 2.8e-2, **S Fig 5C**). Similar to the results in IPF, T-GEN-identified genes in COPD did not show any

enrichment in cell types of the lung. Nevertheless, these results support the cWAS results indicating a role for endothelial cells in COPD.

We further validated the findings on IPF-myofibroblast and COPD-endothelial associations using a lung scRNA-seq dataset with both diseased and healthy individuals [33]. This recent study profiled 32 IPF patients, 18 COPD patients, and 28 controls, and we compared major cell type proportions across these three groups of samples (**S6A Fig**). In IPF patients, the myofibroblast cell type proportion was significantly higher than that in healthy individuals (p = 1.3e-3, **Fig 7A**) compared with other major cell types (**Fig 7B**). We also conducted pathway analysis on both up- and down-regulated genes in IPF myofibroblast cells (**Fig 7C**). The top enriched pathways of upregulated genes mostly function as the extracellular matrix [41] (ECM), a network playing an important role in cell adhesion and linking glycoproteins with fibrous proteins, supporting the importance of the fibroblast-to-myofibroblast migration process in IPF. In COPD, despite low endothelial cell counts and the limited sample size in the single cell data (**S6B Fig**), analysis of upregulated genes in COPD endothelial cells (**Figs 7D and S6B–S6D**) suggests the involvement of DNA-binding transcription activity and higher activity of COPD endothelial cells compared to control endothelial cells.

## Discussion

Recent analyses have devoted great efforts to understand GWAS findings for traits and diseases. Several methods have been developed to link identified variants to genes based on genomic locations [2], epigenetic annotations, or eQTL regulations [21]. At the cell type or tissue level, methods like LD score regression [42] and FUMA [43] utilize annotation information or expression data to investigate the genetic enrichment pattern in cell types or tissues. Differing from previous methods, cWAS is a novel statistical framework to interpret GWAS findings in a cell type proportion manner. It helps researchers gain insights into the relationship between cell type GRPs and diseases. In both simulation and real-data analyses, cWAS has shown a higher statistical power and identified more significant results than MAGMA (implemented in FUMA) and LD score regression (**S1 Text**). cWAS is complementary to cell type-disease associations identified solely through genetic association or heritability enrichment, especially when genetic signals are mediated by regulating cell type proportions. Identified disease-associated cell type proportions can potentially serve as biological markers in clinical practices to identify individuals with higher genetic risk [44,45].

Applying cWAS to GWAS summary statistics from 56 traits, we found that previously genetically correlated traits also have correlated associations with GRPs of cell types. Applications of cWAS to breast cancer, IPF, COPD and Type-I diabetes [46–48] (**S1 Text**) identified cell type proportion-trait associations, which were supported by either previous findings or our analysis of other data. Specifically, a high proportion of CD8[+] T cells was identified as protective in breast cancer development based on both transcriptome and cWAS analyses. Survival analysis using imputed GRPs of cell types also implied a protective effect of higher CD8[+] T cell proportion in breast cancer prognosis. These findings support the importance of CD8[+] T cell proportions for both disease onset and prognosis in breast cancer.

We noted that transcriptome analyses of breast cancer patients have also identified the importance of CD8[+] T cells. Utilizing breast tumor infiltration data, multiple published survival studies [49, 50] found protective effects of high CD8[+] T cell proportions in the tumor tissue for breast cancer. In contrast to TCGA results based on the observed breast tissue expression data, cWAS identified genetically regulated cell type proportions in whole blood, which are more likely to cause the development of the disease instead of being affected by the disease onset. Although the mechanisms in the prognosis and development of breast cancer

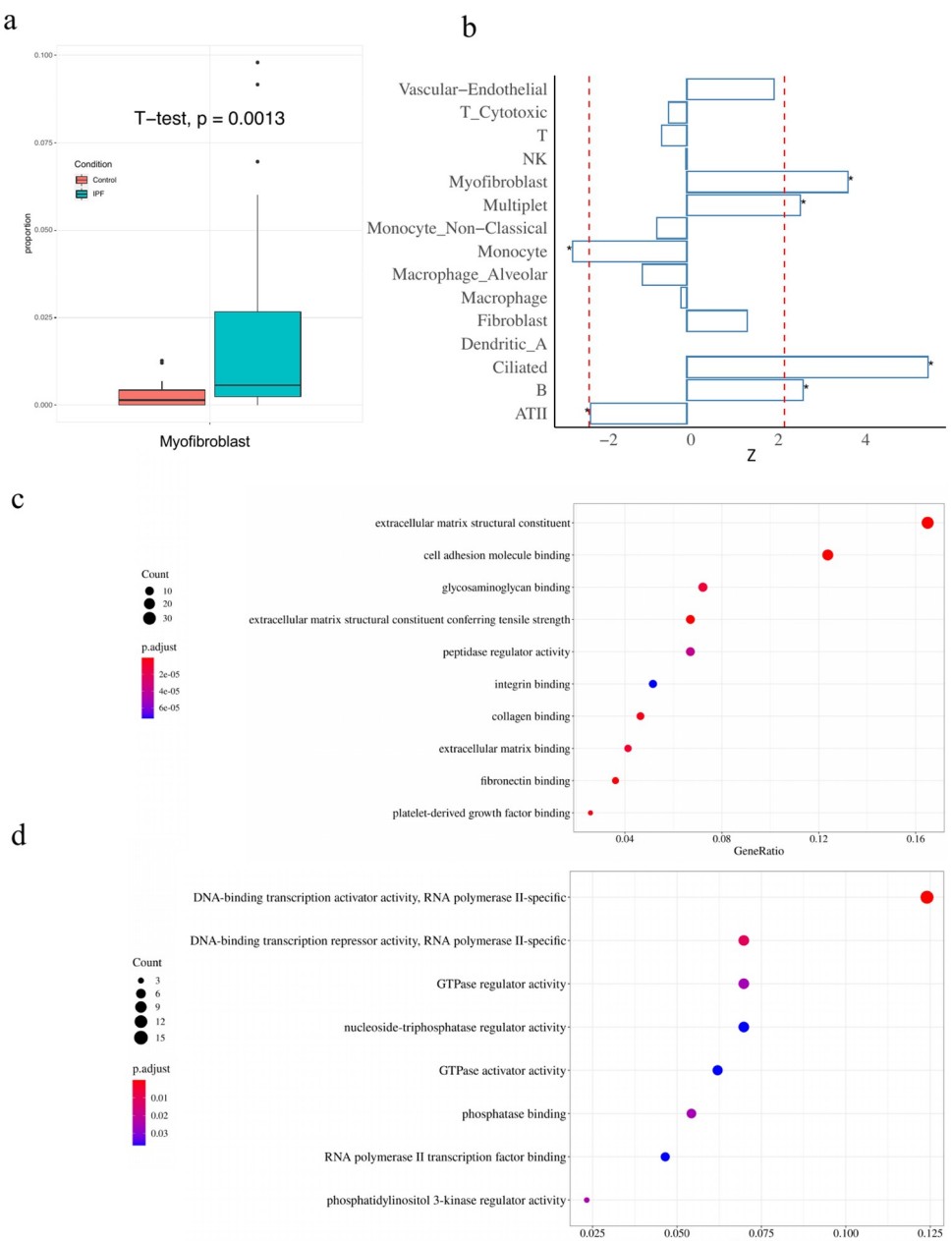

**Fig 7. IPF myofibroblast and COPD endothelial cell type proportion validation in the separate scRNA-seq atlas.**
a) Boxplots of myofibroblast cell type proportions in 32 IPF patients and 28 controls. The vertical axis is the cell type proportion of myofibroblast. The IPF myofibroblast cell proportion is significantly higher than that in controls with p-value = 1.3e-3 by t-test. b) Bar plots of z scores when cell type proportions were regressed on conditions of IPF and control. The red line indicates the significance threshold (0.05) after the Bonferroni correction. The star indicates the significant cell types after Bonferroni correlation. All the cell types with z scores greater than 2 are labeled with an asterisk. Only cell types whose proportions are more than 1% are shown. Myofibroblast ranks second in these 15 major cell populations. This difference may be related to the genetically mediated regulation of cell type proportion based on the cWAS results. c) Dotplot of Gene Set Enrichment Analysis (GSEA) results of IPF myofibroblast up-regulated genes. The dot size is the gene counts found in the pathway. The colors indicate the hypergeometric test p-values. Most top enriched pathways are related to ECM and cell adhesion. d) Dotplot of GSEA results on COPD endothelial up-regulated genes. The dot size is the gene counts found in the pathway. The colors indicate the hypergeometric test p-values. The pathways indicate a stronger DNA-binding transcription activity.

are not necessarily the same, the converging evidence from different approaches used here suggests the importance of CD8[+] T cells in breast cancer.

We also note that previous COPD research has already implied the importance of endothelial cells [51], which are involved in both the initiation and progression of COPD as well as other lung diseases, such as asthma and emphysema. More specifically, endothelial cells play a role in the transendothelial migration (TEM), through which neutrophils move to lung tissue and respond to the residential inflammation [52]. Additionally, endothelial apoptosis in the lung initiates and contributes to the progression of COPD disease [51,53]. Previous genetic research also identified the importance of endothelial cells in COPD [40] using ATAC-seq data and emphysema [54]. This further lends support to the involvement of endothelial cells in developing COPD.

Similar to many statistical methods, cWAS is also highly dependent on the data used, more specifically, the single cell data. The single cell data used in signature gene expression curation can affect cWAS performance, including the cell types included and the signature gene expression levels. More single cell databases with larger sample sizes, higher resolution and more comparable experiment pipelines across more tissues will aid in its further applications and result interpretations. To mitigate the batch effects across various tissues, for trait-trait correlation analysis and multi-tissue association analysis, we used the HCL dataset to extract signature gene matrices since the test results will be comparable across tissues due to relatively small batch effects and the same cell labeling criteria in all tissues. For breast cancer, we used the LM22 matrix, which was curated based on the Affymetrix microarray data, to extract the signature matrix in whole blood. In IPF and COPD, the signature matrix was curated using single cell data of lung from HCL, which consists of 23,878 cells from 20 cell types. Notably, due to effects of sample obtaining process (like tissue positions) in experiments, cell type composition in lung single cell data can be strongly biased and only reflect the local cell type compositions, with 90% of the cells being immune cells. Despite this limitation, we identified a non-immune cell population in COPD. Our results based on fetal brain single cell data relied on the assumption that the genetic regulation of gene expression is the same in both adult and fetal tissues. The assumption could be violated for tissues still undergoing development in fetuses [55]. The accuracy of cWAS results could be further improved if matched genotype and cell type proportion data are available for identifying cell type proportion QTLs.

In cWAS, we assumed that some genetic effects on disease are mediated through affecting cell type proportions. It is difficult to estimate the effects of genetic variants on cell type proportions due to the limited sample sizes in single-cell studies. Thus, we integrated the eQTL information of signature genes here to approximate the genetic effects on cell type proportions. We note that we do not make any assumptions on whether genotypes affect cell type proportions by regulating gene expression or the other way around (Section "Comparisons of different modeling assumptions for genetic effects on bulk expression" in the **S1 Text**). We also compared different modeling assumptions of genetic effects on bulk expression and found that with current datasets of limited sizes, our cWAS framework can better predict gene expression than the cell-type-specific eQTL approach (lines 239–256 in the **S1 Text**). Moreover, we assumed that the cell-type-specific signature gene expression levels are similar across individuals, which is an assumption commonly used in cell type deconvolution methods [20,32]. We have also evaluated the impacts of our assumptions when focused on the genetic components of cell type proportions. More specifically, we found that cis-eQTLs of those genes are less likely to be cell-type-specific eQTL of those genes (p = 1.9e-307), which further suggests that the cis-eQTL effects incorporated in cWAS models are less likely to affect cell-type-specific signature gene expression levels. Further explorations of considering individual variations in both signature gene expression levels and cell type proportions can potentially

improve the cell type deconvolution performance. Besides, cWAS can be further extended in different directions. First, considering the differentiation trajectory between cell types will further pinpoint the associated cell types or even causal cell types, although not all differentiation trajectories are known in human tissues. Second, when analyzing specific traits across tissues to identify the most signal-enriched tissue, we found that traits like BMI and height are associated with cell types in almost all tissues, even though both BMI (p = 1.4e-13) and height (p<2e-16) had the strongest signals in whole blood tissue. This may be due to the complex biological processes involved in these traits. Future work can explore the potential of jointly modeling multiple traits to identify trait-specific associations with cell type proportions. Third, unlike many other cell type deconvolution methods, we did not impose the non-negativity constraint nor requiring the cell type proportions sum to 1. Utilizing GWAS summary statistics, it is technically intractable to get the closed form of a test statistic to quantify the effects of imputed cell type proportion under these constraints. The imputed expression levels were adjusted for covariates like age and sex. Therefore, the non-negative constraint is impractical since the imputed expression is not always positive. Also, empirically we found that the CIBERSORT estimated cell type proportions (with both non-negative and sum-to-1 constraints) showed significant correlation with estimated cell type proportions from simple linear regression (without either constraint) in 78.6% of cell types (mean correlation: 0.36). Utilizing the oneK1k dataset (N = 982) in peripheral blood, we found that for 4 out of 20 cell types, imputed genetically regulated cell type proportions explained significant amount of cell type proportion variances (p value <0.1). Even though the significance level is not that high, it can be partially due to limited sample sizes and limited cis-expression heritability (mean $h^2$ of 0.1 in previous studies). However, an alternative way to tackle those difficulties in the future will help to gain better biological insights. We found that the number of cWAS-identified associations is significantly correlated with the number of genes used in cWAS analysis (p = 6.67e-3, cor = 0.14, **Table J in S1 Table**), which means that more genes utilized may lead to higher statistical power, but also higher type-I error. Here we only consider signature genes in cWAS analysis to serve as a conservative approach to identify disease-associated genetically regulated cell type proportions. Previous methods consider all genes for cell type proportion decomposition. Integrating all genes will also lead to potential random effects, which are mitigated by considering cross-subject variance of genes (MuSiC [56]) or multiple datasets (SCDC [57], InteRD [58]). Further extension of cWAS in considering more genes worth exploration.

To conclude, different from bulk RNA-seq or scRNA-seq analysis comparing patients and healthy individuals, cWAS assesses the association between GRPs of cell types and diseases, and the inferred results better imply the causal effect of cell type proportions on disease outcomes. Compared to commonly used genetic enrichment methods such as MAGMA and LD score regression, cWAS provides a novel way to investigate cell type-disease association. Both simulation and real data analyses have demonstrated the statistical power of cWAS in providing new insights in understanding the genetic etiology of human diseases from the cell type proportion perspective.

## Online Methods

### Expression imputation model training

Tissue-specific expression imputation models were trained in 44 tissues using matched individual-level RNA-seq and whole-genome sequencing data from the GTEx (v8) project. We focused on common SNPs (minor allele frequencies > 0.05) by filtering out SNPs whose allele frequencies were smaller than 0.05. RNA-seq data were adjusted for possible confounding factors, including the first five genotype principal components (PCs) and different numbers of

Probabilistic Estimation of Expression Residuals (PEER) factors. Only cis-SNPs located within 1Mb from the transcription starting site of each gene were considered for training the gene expression imputation model.

Ten-fold cross-validated elastic-net models were applied to build gene expression imputation models, with the parameter $\alpha$ as 0.5 and the optimal $\lambda$ selected via the function cv.glmnet provided in the 'glmnet' package. Only gene expression imputation models with FDR < 0.05 were considered in the following analysis. To make the test results more robust, we only considered those models with an imputation accuracy higher than the median level in each tissue.

### Single cell datasets preprocessing

All single-cell data used in this project were obtained from public repositories. In the trait association analysis, we obtained the tissue-specific signature matrices from the Human Cell Landscape (HCL) [25], sequenced on the microwell-seq platform. HCL provides a coherent sequencing procedure that can minimize the batch-effects to have a higher consistency, making the trait-trait correlation analysis feasible. To better utilize HCL, we manually cleaned the cell type annotations across the tissues to have a consistent cell type naming rule. Because several cell types had a limited number of observed single cells, we only kept the cell types with more than 50 observed single cells.

We found that the curation of signature matrices might not be representative enough if they were only based on the raw counts due to the high drop-out rate of single cell data. To alleviate this problem, we applied SAVERX [59], a deep Bayesian autoencoder single cell imputation tool implemented with transfer learning, on the single cell expression profile to impute drop-out events before signature matrix computation. SAVERX may distinguish the dropout and real zero expression, which helps to get a more accurate cell type-specific average expression. It is common when some single cell datasets have a rare cell population. The limited cell counts make the average expression profile across cells unstable for signature matrix. Therefore, we filtered out cell types with low counts and only kept the major cell types.

In lung disease analysis, to get the signature matrix with deeper sequencing depth and more accurate cell type annotations, we used control samples in the IPF cell atlas [30], which contains 312,928 cells from subjects with IPF and without IPF. As it is not allowed to denoise a large scRNA-seq profile with more than 20,000 cells in SAVERX due to computational limitations and the signature gene list is robust to the size of a randomly sampled subset in lung atlas, we partitioned the lung atlas randomly to get a smaller subset with 20,000 cells. For the signature matrix with more cell types, we include all observed cell types with cell counts larger than 100, while the signature matrix curated for the original two GWAS summary statistics only included the main 20 main cell types annotated in the IPF cell atlas.

### Association analysis

After getting SNP weights $\hat{\beta}_t$ on the tissue-specific gene expression imputation models, we further combined them with published GWAS summary statistics to estimate cell-type associations with disease phenotypes. For a specific cell type $c$, we model the association between a phenotype $Y$ and its genetically regulated cell type proportions $\hat{F}_c$ as $Y = \hat{F}_{(.,c),t}\gamma_c + \eta$. From the linear deconvolution of genetically imputed tissue-specific gene expression, we can estimate the genetically regulated cell type proportion as follows:

$$\hat{F}_t = \hat{B}_t S_t (S_t^T S_t)^{-1} = X\hat{\beta}_t S_t (S_t^T S_t)^{-1}$$

where $\hat{F}_t$ is the cell type proportion matrix in tissue $t$, $S_t$ is the expression matrix of cell-type

specific signature genes, $\hat{B}_t$ is the imputed gene expression level matrix for all signature genes in tissue $t$, $\hat{\beta}_t$ is the SNP weights in the gene expression imputation model for tissue $t$, and $X$ is the genotype matrix.

The full model of genetic effects on trait is:

$$Y = X\omega + \eta_1$$

$$= X(\omega_{prop} + \omega_{non}) + \eta_1$$

$$= \sum_t X\beta_t S_t (S_t^T S_t)^{-1} \gamma_t + X\omega_{non} + \eta_1$$

$$= \sum_t \hat{F}_t \gamma_t + X\omega_{non} + \eta_1$$

where $Y$ is the trait value, $X$ is the genotype matrix, $\omega$ is the GWAS effect size vector for the SNPs, $\omega_{prop}$ represents the genetic effects mediated by regulating cell type proportions, $\omega_{non}$ represents those not mediated by cell type proportions, $\beta_t$ represents the eQTL effects on regulating tissue-level expression of signature genes in tissue $t$, $\gamma_t$ is the effect size vector of genetically-regulated cell type proportions for tissue $t$ on the trait of interest, and $S_t$ is the signature gene expression matrix in tissue $t$. In our set-up, we use $X\beta_t S_t (S_t^T S_t)^{-1}$ to infer the genetically regulated cell type proportions $\hat{F}_t$ in tissue t, which is represented by $\hat{F}_t$. In the model, we assume that the genetically regulated cell type proportion effects on trait value $\gamma_t$ in each tissue are unrelated with each other, and that the effects mediated by cell type proportions $\omega_{prop}$ are unrelated with $\omega_{non}$.

When the individual level data are not available, we cannot obtain the cell proportions $\hat{F}_t$. By considering the genotype-phenotype association $Y = X\omega + \eta_1$, where $\omega$ stands for GWAS effect sizes and $\eta_1$ is the residual effect, we can indirectly estimate the coefficient $\gamma_c$ as follows:

$$\hat{\gamma}_c = \frac{cov(Y, \hat{F}_{(.,c),t})}{var(\hat{F}_{(.,c),t})} = \frac{cov(X\omega + \eta_1, X\hat{\beta}_t S_t (S_t^T S_t)^{-1})}{var(\hat{F}_{(.,c),t})}$$

$$= \frac{cov(X\omega, X\hat{\beta}_t S_t (S_t^T S_t)^{-1}) + cov(\eta_1, X\hat{\beta}_t S_t (S_t^T S_t)^{-1})}{var(\hat{F}_{(.,c),t})}$$

$$= \frac{E(\omega^T X^T X\hat{\beta}_t S_t (S_t^T S_t)^{-1}) - E(X\omega)E(X\hat{\beta}_t S_t (S_t^T S_t)^{-1})}{var(\hat{F}_{(.,c),t})}$$

$$= \frac{E(\omega^T X^T X\hat{\beta}_t S_t (S_t^T S_t)^{-1})}{var(\hat{F}_{(.,c),t})}$$

$$= \sum_p \frac{var(X_p)\omega_p M_{c,p}}{var(\hat{F}_{(.,c),t})}$$

where $A = (S_t^T S_t)^{-1}$, $M_c = \hat{\beta}_t S_t A_c$, $M_{c,p}$ is the element in the $M$ vector for the $p$th SNP, and we assume that the GWAS residual effect $\eta_1$ is independent of gene expression imputation weights, the mean of $\omega$ is 0 and the mean of $\hat{\beta}_t$ is 0.

To further get the z-score statistic for each cell type $z_c = \frac{\hat{\gamma}_c}{se(\hat{\gamma}_c)}$, we would need to get the variance of the estimated coefficients $\hat{\gamma}_c$. Based on simple linear regression, we have:

$$var(\hat{\gamma}_c) = \frac{var(\eta)}{n \times var(\hat{F}_{(.,c),t})} = \frac{var(Y)(1 - R_c^2)}{n \times var(\hat{F}_{(.,c),t})}$$

where $R_c^2$ is the squared correlation between the phenotype $Y$ and the predictor $\hat{F}_{(.,c),t}$. At the same time, based on the phenotype-genotype association from GWAS and assuming the additive genetic effects in the GWAS model and the genetic effects are independent of residual effect, we have:

$$var\left(\omega_p\right) = \frac{var(\eta_1)}{n \times var(X_p)} = \frac{var(Y)(1 - R_p^2)}{n \times var(X_p)}$$

where $R_p^2$ is the squared correlation between the phenotype $Y$ and the predictor $\hat{X}_p$. Combining the equations above, we can get the $z_c$ statistic formulated as follows by assuming that a single SNP or a single cell type can only explain a small percentage of phenotype variance:

$$z_c = \sum_p \frac{var(X_p)\omega_p M_{c,p}}{var(\hat{F}_c)} / se(\hat{\gamma}_c)$$

$$= \sum_p \frac{var(X_p)\omega_p M_{c,p}}{var(\hat{F}_c)} \sqrt{\frac{var(\hat{F}_c) \times n}{var(Y)(1 - R_c^2)}}$$

$$= \sum_p \frac{var(X_p)\omega_p M_{c,p}}{var(\hat{F}_c)} \sqrt{\frac{var(\hat{F}_c)(1 - R_p^2)}{var(X_p)var(\hat{\omega}_p)(1 - R_c^2)}}$$

$$\approx \sum_p sd(X_p) z_p M_{c,p} / sd(\hat{F}_c)$$

and $z_p$ is the z-score for SNP $p$ for GWAS summary statistics for the phenotype of interest. To make the final test results more stable and based on empirical experience, we require that we have 50 or more signature genes. Admittedly, we note that we only use cis-eQTL effects due to the limited power in identifying and estimating trans-eQTL effects and assume that signature gene expression levels are similar across individuals. The validity of these assumptions is more comprehensively evaluated in the section "Comparisons of different modeling assumptions for genetic effects on bulk expression" in the supplementary text (S1 Text).

## Simulation

In simulation studies, we randomly sampled 10,000 individuals from the UK Biobank dataset. Based on their genotypes of common SNPs and gene expression imputation weights trained above, we imputed their genetically regulated gene expression levels in whole blood and lung. Based on the LM22 signature matrix and simple linear regression, we imputed the cell type proportions for each sample in whole blood and used the signature matrix curated from the HCL database to get the cell type proportions for lung tissue. For power analysis, we simulated phenotypes based on the imputed cell type proportion of M1 macrophages from whole blood tissue under different cis-eQTL heritability values from 0.01 to 0.09 by assuming the effect size

of each cis-SNP follows the same normal distribution. Here we defined the heritability as the phenotypic variance contributed by the imputed cell type proportion of M1 macrophages. Then we used PLINK to conduct GWAS analysis to obtain the GWAS summary results. Sex and first ten principal components of genotypes were adjusted. These GWAS summary results were used in the cWAS test to identify disease-cell type proportion association in whole blood. For the type-I error analysis, the disease phenotypes were simulated based on imputed proportions of basal cells in lung tissue. Similar to the analysis in the whole blood tissue, we obtained the GWAS summary statistics but the heritability we considered was 0.05, 0.1, and 0.5. After getting the GWAS results, we applied cWAS to identify disease-cell type proportion associations in whole blood tissue.

## Signature gene expression matrix curation

Only protein-coding genes were considered in our analyses. We selected the signature genes by differential expression (DE) analysis, including Wilcoxon rank sum test, Model-based Analysis of Single-cell Transcriptomics (MAST) [60], and ANOVA. Among these methods, MAST is a DE framework that takes cell size and drop-out rates into consideration. The Wilcoxon Rank Sum test and MAST for DE analysis were conducted by the FindMarkers() command in Seurat (3.1.5). Bonferroni correction at $\alpha = 0.05$ was used. When a large number of DE genes were selected, we kept the DE genes which were upregulated and differential to a single cell population. We took the intersection between significant DE genes and the GTEx-V8 genes of the corresponding tissue and included them in the signature matrix. By setting different thresholds and applying appropriate DE analysis approaches for filtering, we aimed to get the signature matrices. We computed the cell type-specific gene signature matrices by the average expression levels across cells within the cell populations in the final step.

## Survival analysis in TCGA data

The imputation of the tissue-specific bulk RNA-seq expression for the TCGA-BRCA samples was based on individual germline genotypes and corresponding expression weights trained above. We followed the same procedure in the work of Huang et al. to process the germline genotypes [30] from TCGA. Missing SNPs were not considered.

After getting the imputed tissue-level gene expression, we used linear regression to estimate the genetically regulated cell type proportions for each sample. For survival analysis, we compared the disease-free survival times between groups with a high and lower percentage of genetically predicted cell type proportion of the identified cell type. More specifically, we extracted those two groups of samples with extremely high (e.g.: top 10%) or low genetically predicted cell type proportion levels. Then, we compared disease-free survival times for the samples with high genetically regulated cell type proportions and those with low ones.

## Curation of GWAS summary data

We collected GWAS summary data of 56 phenotypes. Their detailed information can be found in **Table B in S1 Table**. We selected studies with most of the populations of European ancestry to reduce bias due to population stratification. GWAS summary statistics were curated by first filtering out SNPs with minor allele frequencies less than 0.05. For datasets without rsID, we used the human genome reference build 37 to map to the corresponding rsID. For datasets without Z score or P-value, we manually calculated Z scores and p-values using other available information such as beta, odds ratio, and standard error. After these steps, all GWAS summary statistics contained rsID, reference and alternative alleles, Z scores, p values, and sample sizes.

For down-sampled GWAS summary statistics, we considered the z scores as $z = \sqrt{n}\beta$, where $\beta$ is the reported effect size in GWAS results and $n$ is the sample size. When we reduced the sample size of the GWAS summary statistics, we consider the $z_{downsample} = \frac{z_{original}}{\sqrt{R}}$, where $R$ is the ratio of the sample size in the original GWAS summary statistics over the sample size in the down sampled GWAS summary statistics. We considered $R$ = 2,3,4, and 5 in our study here.

## MAGMA gene-based analysis

To generate annotations, the gene location files using the human genome reference build 37 were downloaded from the MAGMA [2] software website as the input of the gene location file. The SNP location file was generated by extracting SNPs from curated GWAS summary data and mapping to the genome locations using build 37. Annotations were generated with the command: magma—annotate—snp-loc [SNPLOC_FILE]—gene-loc [GENELOC_FILE]—out [ANNOT_PREFIX]. Next, gene analysis was performed for each phenotype using the annotation files generated from the previous step. European panels of the 1000 Genomes phase 3 data downloaded from the MAGMA software website were used as the reference. The following command was used to generate gene analysis results: magma—bfile [REFDATA]—gene-annot [ANNOT_PREFIX].genes.annot—pval [PVAL_FILE] ncol = N snp-id = SNP pval = P—out [GENE_PREFIX]. To benchmark cWAS with MAGMA [2], we used two sources of single cell data to perform cell type MAGMA gene analysis. One is the 60 pre-processed scRNA-seq datasets provided by MAGMA[URLs, **Table E in S1 Table**]. In addition, we extracted single cell datasets from 14 tissues in HCL, including 12 adult tissues (adipose, artery, esophagus, heart, lung, pancreas, peripheral blood, prostate, spleen, stomach, thyroid, and uterus) and two fetal tissues (brain and skin). In the end, we performed MAGMA gene-property analyses (v1.07) of 56 traits(**Table B in S1 xlsx**) and single cell data using the following command: magma—gene-results [GENE_PREFIX].genes.raw—gene-covar [SCDATA]—model condition-hide = Average direction = greater—out [OUT_PREFIX]. Bonferroni correction at alpha = 0.05 was performed per trait-dataset pair during the gene-property analyses to obtain significantly associated cell types.

## LD score regression

To benchmark cWAS results with LD score regression, we first used cell-type annotations and the Wilcox test to identify cell type-specific differentially expressed (DE) genes in HCL. When a signature gene is the DE gene for multiple cell types simultaneously, we assigned it to the cell type with the largest logFC. Then we coded the "annotation" scores of the SNPs locating within the 1kb of up/downstream 100kb of DE genes as 1 and the rest as 0 for each cell type as LD score regression annotation input. We evaluated the heritability enrichment of disease GWAS signals on these cell type annotations accounting for baseline annotations. For comparison of cWAS and epigenetic annotations, we used ATAC-seq data GSE165659. We utilized FindAllMarkers function in Seurat to perform DE peaks discovery and recognizes positive markers for each cell type by linear regression. All the DE peaks for each cell type should have a log fold change (logFC) greater than 0.1 and be present in at least 15% of the overall cell population. We also found that the DE peak selection is robust to the choices of minimal logFC and minimal presence fraction. We coded 500kb up/downstream of the ATAC-seq signal as 1 and the rest genome as 0 for each cell type and tested for the heritability enrichment of disease GWAS signals on cell type epigenetic annotations accounting for baseline annotations.

### T-GEN and cell-type enrichment analysis

For GWAS summary statistics of BC, COPD, and IPF, we utilized the pre-trained T-GEN imputation models (v6) in https://github.com/vivid-/T-GEN, which is a TWAS-like method incorporating epigenetic annotation to build gene expression imputation models, and then identified significant trait-associated genes in each tissue. We considered 26 tissues in total.

For each tissue of interest, we utilized the corresponding single cell dataset in the control samples to link genes to cell types. The single cell dataset in PBMC used here is from Unterman et al [61]. For each cell type, we standardized the gene expression levels across all genes to make the mean of the overall distribution to be 0 and the standard deviation to be 1. Then each gene was assigned to the cell type where this gene had the largest standardized expression levels across all cell types by assuming the relative expression extremity in each cell type represents cell-type specificity. Then the binomial test was conducted to test if those T-GEN identified genes are enriched in specific cell types, and the Bonferroni correction was applied for multiple test correction.

### Trait-cell type association analysis across tissues

Using the signature gene matrix processed from the HCL database, we applied cWAS to obtain cell type association results for each trait across different tissues. To investigate trait-trait correlations, we considered the test-statistics (z scores) of all cell types in a trait as the representation vector of the trait. Then for any two traits, we calculated the Pearson correlation between their corresponding two z score vectors and associated p-value to quantify the similarity between these two traits with respect to cell type associations. Similarly, to consider the correlation of the effects of a shared cell type between tissues, for a specific cell type, we treat its association z scores with all traits in one tissue as a vector $v_1$. We then put its association z scores with all traits in a second tissue as a vector $v_2$. To study the tissue-tissue correlation for the shared cell type effects, we calculated the Pearson correlation between $v_1$ and $v_2$.

In the across-tissue analysis, for each trait, we firstly identified the significant cell type associations after the Bonferroni correction in each tissue. Then across all tissues, we identified the most significant cell type association signals.

### Differentially expressed genes from bulk and cell type enrichment analysis

Differentially expressed (DE) gene lists in IPF and COPD patients were downloaded from previous publications [39,62]. We curated the cell-type specific gene expression matrix in lung tissue using the published single cell data. Then for each DE gene, we identified the cell type having the highest expression of this gene. Then for each cell type, we calculated the enrichment of upregulated IPF and COPD DE genes compared to other genes in this cell type. The binomial test was used to calculate the significance level of the enrichment pattern and then Bonferroni correction was further applied to select the significant cell types.

The Gene Set Enrichment Analysis was conducted by the gseGO() command in Bioconductor package clusterProfiler (3.14.3). All the parameters were set to the default values, where Benjamin–Hochberg correction at $\alpha = 0.05$ was used as the cutoff.

### URLs

Human cell landscape: http://bis.zju.edu.cn/HCL/; GEO GSE134355 https://www.ncbi.nlm.nih.gov/geo/query/acc.cgi?acc=GSE134355; CNGBdb CNP0000325 https://db.cngb.org/search/project/CNP0000325/

GTEx data: https://gtexportal.org/home/; dbGaP Accession phs000424.v8.p2

Roadmap Epigenomics project: http://egg2.wustl.edu/roadmap/data/byFileType/signal/consolidated/

BC summary stats: http://bcac.ccge.medschl.cam.ac.uk/; https://bcac.ccge.medschl.cam.ac.uk/bcacdata/oncoarray/oncoarray-and-combined-summary-result/gwas-summary-associations-breast-cancer-risk-2020/;

COPD summary stats: https://pubmed.ncbi.nlm.nih.gov/24621683/; https://pubmed.ncbi.nlm.nih.gov/30804561/

IPF summary stats: https://github.com/genomicsITER/PFgenetics

MAGMA: https://github.com/Kyoko-wtnb/FUMA_scRNA_data

OneK1k: https://onek1k.org

## Supporting information

**S1 Fig. The workflow of curating gene expression signature matrix in each tissue.** Single cell data across multiple cell types in a given tissue are firstly imputed by SAVER-X and then significant differentially expressed (DE) genes are identified based on cell-type level DE analysis. Finally, for those identified DE genes, their average gene expression levels are computed within each cell type.
(TIFF)

**S2 Fig. Cell type-trait associations across 56 traits identified by cWAS. I**n 36 tissues, the significant/most-significant associated cell type results are shown in the figure. Blue colors indicate the negative correlations between traits and the corresponding associated cell type proportions while red colors indicate the positive correlations.
(TIFF)

**S3 Fig. Cell type expression pattern of breast cancer-associated genes identified by TWAS analysis. a**) As in previous figures, the star indicates the significant cell types after Bonferroni correction in whole blood. The fold indicates (x axis) the enrichment level of breast cancer-associated genes among those genes with high expression specificity in the corresponding cell type. b) Expression levels of identified breast cancer-associated genes in different cell types of whole blood. Red diamonds indicate the mean expression level of breast cancer-associated genes in corresponding cell types.
(TIFF)

**S4 Fig. Survival analysis results of TCGA breast cancer patients.** We considered cell type proportions estimated from the assayed expression levels in tumor tissues of TCGA breast cancer patients. a) In patients of European ancestry with basal breast cancer, we compared the survival distributions of patients having the top 10% and bottom 10% of estimated $CD8^+$ T cells proportions. b) For Luminal A patients with European ancestry, we compared patients with the top 10% and bottom 10% of estimated $CD8^+$ T cells proportions. c) For luminal B patients of European ancestry, we compared patients with the top 10% and bottom 10% of estimated $CD8^+$ T cell proportions. We considered different percentages of patients in these comparisons to demonstrate the most significant discriminations between selected two groups of individuals.
(TIFF)

**S5 Fig. MAGMA analysis results of IPF and COPD GWAS summary statistics.** In all figures, the vertical grey dashed lines indicate the significance threshold after Bonferroni correction. The red bars indicate the corresponding cell types of interest in IPF and COPD. a) Bar plots of MAGMA cell type association results between IPF and all cell types of the MAGMA

processed GSE93374_Mouse_Arc_ME_level2 dataset [63]. Fibroblast-related cell types are highlighted in red. b) Bar plots of MAGMA cell type association results between COPD and cell types from lung tissue of the MAGMA processed TabulaMuris_FACS_all dataset [64]. Endothelial-related cell types are highlighted in red. c) Bar plots of MAGMA cell type association results between COPD and all cell types from the MAGMA processed GSE99235_Mouse_Lung_Vascular dataset. Endothelial-related cell types are highlighted in red.
(TIFF)

**S6 Fig. IPF myofibroblast and COPD endothelial cell type proportion validation in the separate scRNA-seq atlas.** a) Boxplots of cell-type proportions comparisons across IPF, COPD, and controls in lung tissue. The horizontal axis represents the major cell types. The vertical axis is the cell type proportions. The immune cells are the majority of the data. The cell type proportion has a non-negligible variance across different conditions. b) Boxplots of two endothelial subtype proportions comparison between COPD and controls. The vertical axis represents cell-type proportions. To compare the cell type proportion distributions between COPD and controls, we conducted a t-test which was not significant. However, the direction is consistent with cWAS finding for vascular endothelial cells. We still consider these results inconclusive due to the low endothelial cell counts. c) Dot plots of GSEA on IPF myofibroblast down-regulated genes. The dot size is the gene counts found in the pathway. The colors indicate the hypergeometric test p-values. d) Dot plots of GSEA on COPD endothelial down-regulated genes. The dot size and colors are the same as in b).
(TIFF)

**S7 Fig. Impacts of known signature gene fractions and disease heritability on cWAS performance using a) the original test; b) the permutation-like test.** FIR: false identification rate, the rate of cell types which are not disease-associated but identified by cWAS; power: the rate of the true disease-associated cell type being identified as disease-associated; top rate: the rate of the cWAS-identified most significant cell types being the true signal cell type. Note that different from Fig 2, we had 300 replicates for each simulation setting instead of 600 replicates here. a) the results of the original test in the main text; b) the results of the proposed alternative permutation-like test.
(TIFF)

**S8 Fig. Impacts of (a) cell type proportions and (b) the number of signature genes on the imputation accuracy of cell type proportion.** Sig: those cell types whose cell type proportion were imputed with p values <0.05 after multiple test correction. Non-sig: cell types that did not pass the significance level after multiple test correction.
(TIFF)

**S9 Fig. Significance levels of correlations between cWAS false discovery rate and estimated cell type proportions (cor_W_prop), and cell type proportion imputation accuracies (cor_W_r2).** "cor_W_prop" indicates the correlation between the false identification rate of each cell type and its proportion. "cor_W_r2" indicates the correlation between the false identification rate of each cell type and its proportion imputation accuracy. "Fraction" is the fraction of known signature genes, "h2" indicates the percentage of disease phenotype variance explained by genetic-regulated cell type proportions.
(TIFF)

**S10 Fig. cWAS model performance as a function of (a) GWAS and (b) eQTL sample size.** Here we assumed all signature gene are known and the proportion of disease phenotype variance explained by genetic-regulated cell type proportions is 10%. The results of each setting

are based on 300 replicates.
(TIFF)

**S11 Fig. Effects of expression similarities between cell types on cWAS performance.** a)
The performance of the original test. b) The performance of the alternative permutation-like
test.
(TIFF)

**S12 Fig. Comparing different assumptions modeling genetic effects on bulk expression
imputation.** "cWAS" indicates the imputation model under the cWAS assumption. "ieQTL_-
CiberSort" indicates that the cell-type specific eQTL effect model and the cell type proportion
were estimated using CIBERSORT; "ieQTL_nonConst" is similar to the "ieQTL_CiberSort",
but its cell type proportions were estimated using linear regression models without constraints;
"all" indicates gene expression modeling considering both cell-type specific eQTL effect and
genetic-regulated cell type proportions.
(TIFF)

**S13 Fig. Comparison of tissue level trait associations identified by cWAS and LD score
regression.** Only traits/tissues with available association results in both methods are plotted.
Darker shade represents smaller p-value. Significant associations after Bonferroni correction
at alpha = 0.05 are indicated using asterisk.
(TIFF)

**S14 Fig. Comparison of tissue level trait associations identified by cWAS and MAGMA
gene analysis.** Only traits/tissues with available association results in both methods are plotted.
Darker shade represents smaller p-value. Significant associations after Bonferroni correction
at alpha = 0.05 are indicated using asterisk.
(TIFF)

**S15 Fig. QQ-plot of cWAS, MAGMA gene analysis, and LD score regression.** QQ-plot test
results for cell type-trait associations across 56 traits. The black line indicates y = x.
(TIFF)

**S16 Fig. Comparison between cWAS and LD score regression using epigenetic annotation.**
IPF and COPD cell type associations identified by cWAS and LD score regression with epige-
netic annotation. Darker shade represents smaller p-value. Significant associations after Bon-
ferroni correction at alpha = 0.05 are indicated using asterisk.
(TIFF)

**S1 Table.** Table A. Statistical power and type I error of FUMA in simulation study. Table B.
Cohort information of GWAS summary statistics. Table C. HCL tissues used in the analysis.
Table D. Shared cell types (mainly immune cells) in all tissues. Table E. cWAS test results in all
HCL tissues for 56 traits. Table F. scRNA-seq datasets used for MAGMA analysis. Table G.
The fraction of cell type proportion explained by genetics. Table H. Significant associations
identified by MAGMA gene analysis. Table I. Significant associations identified by LD score
regression. Table J. Number of significant associations identified by cWAS when utilizing dif-
ferent number of signature genes.
(XLSX)

**S1 Text. Detailed discussion about cWAS results and limitations.**
(DOCX)

## Author Contributions

**Conceptualization:** Wei Liu, Wenxuan Deng, Ming Chen, Hongyu Zhao.

**Data curation:** Wei Liu, Wenxuan Deng, Ming Chen.

**Formal analysis:** Wei Liu, Wenxuan Deng, Ming Chen, Zihan Dong, Biqing Zhu, Zhaolong Yu, Daiwei Tang, Chen Lin.

**Funding acquisition:** Michael H. Cho, Naftali Kaminski, Hongyu Zhao.

**Investigation:** Maor Sauler, Louise V. Wain, Michael H. Cho, Naftali Kaminski, Hongyu Zhao.

**Methodology:** Wei Liu, Wenxuan Deng, Ming Chen, Hongyu Zhao.

**Software:** Wei Liu.

**Supervision:** Hongyu Zhao.

**Writing – original draft:** Wei Liu, Wenxuan Deng, Ming Chen.

**Writing – review & editing:** Wei Liu, Wenxuan Deng, Ming Chen, Chen Lin, Hongyu Zhao.

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
