## [Decision Letter · Decision Letter 0]

19 Feb 2023

Dear Dr Zhao,

Thank you very much for submitting your Methods entitled 'A Statistical Framework to Identify Cell Types Whose Genetically Regulated Proportions are Associated with Complex Diseases' to PLOS Genetics.

The manuscript was fully evaluated at the editorial level and by independent peer reviewers. The reviewers appreciated the attention to an important problem, but raised some substantial concerns about the current manuscript. Based on the reviews, we will not be able to accept this version of the manuscript, but we would be willing to review a much-revised version. We cannot, of course, promise publication at that time.

If you decide to revise the manuscript for further consideration at PLOS Genetics, please aim to resubmit within the next 60 days, unless it will take extra time to address the concerns of the reviewers, in which case we would appreciate an expected resubmission date by email to plosgenetics@plos.org.

We are sorry that we cannot be more positive about your manuscript at this stage. Please do not hesitate to contact us if you have any concerns or questions.

Yours sincerely,

Mingyao Li

Academic Editor

PLOS Genetics

David Balding

Section Editor

PLOS Genetics

Reviewer's **Comments to the Authors:**

Reviewer #1: Liu et al. presented a novel analysis frame cWAS, cell type wide association study that integrates genetic data with transcriptomics data to identify cell types whose genetically regulated proportions (GRPs) are trait-associated. The manuscript is well written and the results are clearly presented. Here are some concerns of mine:

1. How much does the signature matrix affect gene expression? With different signature matrices of the same tissue type, do gene-regulated proportions have the same effect size to phenotypes or the same power?

2. The genetically regulated cell type proportions F ^_c does not reflect the true cell type proportions since (1) it is not normalized with sum-to-one; (2) the imputed bulk expressions are subject to the part that genetic variation can explain. Therefore, the genetically regulated cell type proportions reflect the projection of imputed expression to cell type-specific expression, which genetic variations can explain. Can you discuss more on genetically regulated cell type proportions from imputed gene expression compared to cell type proportions from deconvolution methods?

3. There are deconvolution methods such as MuSiC (Wang et al. 2019), SCDC (Dong et al. 2020), and Bisque (Jew et al. 2020) that utilize all genes in bulk expression without marker gene selection. Aside from the signature matrix and signature genes selected from CIBERSORT, the well-known deconvolution method, can you discuss more why only include the signature genes in the model? What are the pros and cons of including more genes in cWAS study?

Reviewer #2: This article proposes a statistical framework, called cWAS, which interprets GWAS findings in a cell type proportion manner. cWAS has been demonstrated and evaluated in both simulation and real-data analyses with higher statistical power detecting more significant results than existing methods. The identified disease-associated cell type proportions can furthermore serve as biological markers that could be useful to identify individuals with higher genetic risk. The idea of cWAS is interesting, the method is described well, and the data application results are convincing. The followings are some comments that may help further improve the manuscript:

1. More information is needed to describe the simulation study. For example, what does “600” replicates mean? Monte Carlo replicates, or sample size in each Monte Carlo run? It would be good to see how this method performs under different sample sizes. Whether does this method require substantial sample size to achieve desired power? This evaluation will be helpful and can provide guidance in practice.

2. The authors should correct some inconsistent statements throughout the article: for example, on page 7, line 150, it says “When we simulated phenotypes independent of cell type proportions in the whole blood tissue (figure 2c)”. This does not match the information in the description of figure 2 and simulation section on page 25. Are they from blood tissue, or lung tissue? It is confusing.

3. Some notations in formulas are confusing. On page 22, line 478-480, gamma_t is used; however, on page 23, line 494, gamma_c is used. If gamma_c is one element in gamma_t vector, please clearly state it or make it in a more rigorous subscript. Also, on page 5, line 107, the equation is a bit weird and not consistent with the model on page 22. This also raises another concern about applying this method: are effects from all tissues estimated jointly by the model on page 22? any multiple test adjustment applied? More description is needed.

4. On page 22, the estimation of cell type proportion is based on the logic of simple linear model. Any specific reason for considering this simple estimation? How does this estimation ensure the property of sum up equal to one? There are many advanced deconvolution tools developed in recent years. More literature review in this area is highly suggested. Some relevant references are listed below:

“Bulk tissue cell type deconvolution with multi-subject single-cell expression reference (2019)”

“Omnibus and robust deconvolution scheme for bulk RNA sequencing data integrating multiple single-cell reference sets and prior biological knowledge (2022)”

**Have all data underlying the figures and results presented in the manuscript been provided?**

Reviewer #1: Yes

Reviewer #2: Yes

PLOS authors have the option to publish the peer review history of their article (what does this mean?). If published, this will include your full peer review and any attached files.

Reviewer #1: **Yes: **Xuran Wang

Reviewer #2: **Yes: **Chixiang Chen

---

## [Decision Letter · Decision Letter 1]

12 Jun 2023

Dear Dr Zhao,

We are pleased to inform you that your manuscript entitled "A Statistical Framework to Identify Cell Types Whose Genetically Regulated Proportions are Associated with Complex Diseases" has been editorially accepted for publication in PLOS Genetics. Congratulations!

Yours sincerely,

Mingyao Li

Academic Editor

PLOS Genetics

David Balding

Section Editor

PLOS Genetics

Reviewer's **Comments to the Authors:**

Reviewer #1: The authors have answered all my questions.

Reviewer #2: The authors have sufficiently addressed my concerns

**Have all data underlying the figures and results presented in the manuscript been provided?**

Reviewer #1: Yes

Reviewer #2: None

PLOS authors have the option to publish the peer review history of their article (what does this mean?). If published, this will include your full peer review and any attached files.

Reviewer #1: **Yes: **Xuran Wang

Reviewer #2: **Yes: **chixiang chen

**Data Deposition**

http://datadryad.org/submit?journalID=pgenetics&manu=PGENETICS-D-22-01453R1

**Press Queries**

---

## [Editor Report · Acceptance letter]

24 Jul 2023

PGENETICS-D-22-01453R1 

A Statistical Framework to Identify Cell Types Whose Genetically Regulated Proportions are Associated with Complex Diseases 

Dear Dr Zhao, 

We are pleased to inform you that your manuscript entitled "A Statistical Framework to Identify Cell Types Whose Genetically Regulated Proportions are Associated with Complex Diseases" has been formally accepted for publication in PLOS Genetics! Your manuscript is now with our production department and you will be notified of the publication date in due course.

With kind regards,

Jazmin Toth

PLOS Genetics

On behalf of:
